# Myocardial Adaptation in Pseudohypoxia: Signaling and Regulation of mPTP via Mitochondrial Connexin 43 and Cardiolipin

**DOI:** 10.3390/cells8111449

**Published:** 2019-11-17

**Authors:** Miroslav Ferko, Natália Andelová, Barbara Szeiffová Bačová, Magdaléna Jašová

**Affiliations:** Center of Experimental Medicine, Slovak Academy of Sciences, Institute for Heart Research, Dúbravská cesta 9, 841 04 Bratislava, Slovakia; nat.andelova@gmail.com (N.A.); usrdbaca@savba.sk (B.S.B.); jasovam@gmail.com (M.J.)

**Keywords:** cardioprotection, mitochondria, mitochondrial permeability transition pores, mitochondrial connexin 43, cardiolipin

## Abstract

Therapies intended to mitigate cardiovascular complications cannot be applied in practice without detailed knowledge of molecular mechanisms. Mitochondria, as the end-effector of cardioprotection, represent one of the possible therapeutic approaches. The present review provides an overview of factors affecting the regulation processes of mitochondria at the level of mitochondrial permeability transition pores (mPTP) resulting in comprehensive myocardial protection. The regulation of mPTP seems to be an important part of the mechanisms for maintaining the energy equilibrium of the heart under pathological conditions. Mitochondrial connexin 43 is involved in the regulation process by inhibition of mPTP opening. These individual cardioprotective mechanisms can be interconnected in the process of mitochondrial oxidative phosphorylation resulting in the maintenance of adenosine triphosphate (ATP) production. In this context, the degree of mitochondrial membrane fluidity appears to be a key factor in the preservation of ATP synthase rotation required for ATP formation. Moreover, changes in the composition of the cardiolipin’s structure in the mitochondrial membrane can significantly affect the energy system under unfavorable conditions. This review aims to elucidate functional and structural changes of cardiac mitochondria subjected to preconditioning, with an emphasis on signaling pathways leading to mitochondrial energy maintenance during partial oxygen deprivation.

## 1. Introduction

Mitochondria are considered to be one of the most important organelles, not only in terms of their ability to control apoptosis [1] or necrosis [2], but also for their important participation in cardioprotection [3]. Mitochondria can cope with energy demanding situations due to their adaptability. The adaptation mechanisms of mitochondria are very important especially in the heart [4]. Cardiac mitochondria provide more than 90% of the total energy required for the cell [5]. Moreover, mitochondria are able to adapt to new conditions through signaling pathways affecting membrane remodeling, mitochondrial dynamics, or energy production [6,7].

Currently, many studies suggest that regulation of mitochondrial permeability transition pore (mPTP) opening plays a key role in the induction of cardioprotection [8,9,10,11]. Modulation of mitochondrial membrane fluidity through its major component, cardiolipin, or signalization via mitochondrial connexin 43 (mtCx43) leads to myocardial energy maintenance under the conditions of reduced oxygen utilization.

The common denominator of cardioprotection induction seems to be the exposure of the organism to oxygen limiting conditions [12]. The partial or complete absence of oxygen (hypoxia, anoxia) or damage of the respiratory chain affect the changes of biochemical and metabolic processes and induce remodeling of membrane systems [13]. A limited supply or damage in oxygen processing activates signaling pathways that result in structural and functional changes involved in the adaptation of myocardium to pathological conditions.

The mPTP, cardiolipin, and mtCx43 signaling pathways are calcium associated. Calcium (Ca^2+^) ions as major inducers of mPTP opening show a high affinity to cardiolipin [14,15]. The process of hypoxia and subsequent reoxygenation also affect mPTP opening coupled with regulation of Ca^2+^ handling and cardiolipin oxidation [16]. Similarly, mtCx43 forms Ca^2+^ permeable hemichannels allowing Ca^2+^ entry and triggering a permeable transition leading to cell death [17]. In the following parts of this review, we discuss the signaling pathways through mPTP regulation in cooperation with cardiolipin and mtCx43 leading to myocardial adaptation in pseudohypoxia.

## 2. Cardioprotection and Mitochondrial Energetics

Myocardium is highly dependent on sufficient oxygen supply. For this reason, cardiac mitochondria must maintain adequate oxygen to continue oxidative phosphorylation [18,19,20]. Mitochondrial biogenesis is increased at the metabolically active site of the cell where the consumption of adenosine triphosphate (ATP) is increased [21,22]. Therefore, mitochondria occupy up to 35% of the cell volume of cardiomyocytes of heart ventricles [23,24,25]. The oxygen consumption varies depending on the physiological state of the organism [26]. Insufficient oxygen supply, characteristic of pathological situations, is reflected in the reduction of energy production in cardiac mitochondria [22,27,28]. Although cardiac mitochondria are the main energy source of cells, their dysfunction contributes to the development of a wide range of diseases [29,30]. The most common diseases, such as ischemic heart disease [31] or diabetes mellitus [32,33], create conditions in which the organism is exposed to a significant lack of oxygen. Partial (hypoxia) or complete (anoxia) absence of oxygen or the inability to use available oxygen due to damage of the mitochondrial respiratory chain (pseudohypoxia) well characterizes the disease of diabetes mellitus [34] and changes of several biochemical and metabolic processes [32]. Therefore, attention is required to develop new therapeutic approaches directed to mitochondria as target organelles triggering cardioprotection.

The principle of the new cardioprotective models is based on controlled oxygen restriction [35]. One of the first well known phenomena of cardioprotection is ischemic preconditioning (IPC), consisting of several repetitions of short ischemic and subsequent reperfusion episodes that reduce myocardial sensitivity before the next prolonged ischemic episode of the heart [36]. The duration of ischemia is crucial for the rate of myocardial damage [37]. While the early phase of ischemia causes reversible changes of cardiomyocyte and decreases the contractility of myocardium, prolonged ischemia (more than 20 to 30 min) leads to irreversible changes in the metabolism, function, and ultrastructure of the heart [30].

Although many studies have confirmed the efficacy of the classical form of IPC [38,39,40], attention is drawn to an alternative method of controlled induction of short term non-lethal series of ischemic and subsequent reperfusion impulses on specific organs or tissues remote from the heart, known as remote ischemic preconditioning (RPC) [41]. This phenomenon provides protection of myocardium against lethal ischemic damage [42].

An insufficient oxygen supply and nutrients in cardiomyocytes is the main cause of heart ischemia/reperfusion (I/R) injury [43,44]. In a situation with a continuous lack of oxygen, anaerobic glycolysis is preferred [45]. A change in substrate preference used for energy production seems to be the key mechanism favorable for cells with a limited oxygen supply. This is one of the reasons why partial oxygen deprivation is the main factor used in experimental models for the induction of cardioprotection [46].

## 3. Cardiac Mitochondrial Energetics in Partial Oxygen Deprivation

Oxygen deprivation is reflected in specific metabolic changes that result in a balance disorder between fatty acids and glucose oxidation. The restriction of oxygen supply is reflected in changes in preferences for substrates used for energy production [47,48]. In comparison with fatty acid oxidation, a higher amount of ATP is produced by aerobic oxidation of glucose in relation to oxygen consumption [49]. Therefore, glucose is the preferred energy substrate. Despite the fact that fatty acids are less efficient energy substrates compared to glucose, fatty acids are the preferred source of energy in situations associated with impaired mitochondrial function, reduced respiration, and decreased ATP production, such as ischemia of the heart or diabetes mellitus [50].

Increasing oxidation of fatty acids in the heart reduces oxidation of glucose and vice versa. The oxidation of fatty acids increases nicotinamide adenine dinucleotide (NADH) and acetyl-CoA levels, which inhibit pyruvate dehydrogenase (PDH) associated glucose metabolism reduction [51,52]. The process of mutual regulation of glucose and fatty acid metabolism is called the Randle cycle [53]. However, the predominance of fatty acid oxidation during reperfusion versus glucose oxidation negatively affects the activity of the heart [48,54]. Consequently, manipulating heart metabolism to redirect fatty acid oxidation during reperfusion to glucose utilization may constitute a proof-of-concept on how to preserve heart function after ischemia or hypoxia [55,56].

When a sufficient supply of oxygen is ensured, glucose is metabolized by aerobic oxidation [57]. The PDH complex metabolizes glucose to acetyl-CoA, which then enters into the Krebs cycle [58]. A limited supply of oxygen causes phosphorylation of PDH subunits, i.e., PDH inactivation, which is reflected in the inability to metabolize glucose to pyruvate and acetyl-CoA. Then, glucose is metabolized by anaerobic glycolysis to lactate [59,60]. This process is used mainly by cancer cells that are permanently in anaerobic conditions [61]. Oxygen deprivation stimulates the overexpression of hypoxia-inducible factor 1α (HIF-1α) and inactivation of PDH through pyruvate dehydrogenase kinase 1, resulting in the preference of anaerobic glucose oxidation [62,63,64]. This process in which the glucose metabolism is reprogrammed from aerobic to anaerobic is known as the Warburg effect [65]. Despite the fact that ATP production in anaerobic glycolysis is much lower, i.e., two molecules of ATP are produced by anaerobic glycolysis, but up to 36 molecules of ATP by the oxidative phosphorylation of one glucose molecule, anaerobic glycolysis is preferred due to low oxygen consumption [66]. Deletion of HIF-1α affects heart function under normoxic conditions, despite the fact that the heart is protected by HIF-1 against hypoxia [32]. Since PDH and the electron transport chain of mitochondria are the major sources of reactive oxygen species (ROS), a preference for anaerobic glycolysis prevents the apoptosis of cancer cells [67,68]. Moreover, continuous production of ATP is ensured by constant glucose supply (malignancy or hyperglycemia) [69]. In addition, the process of anaerobic glycolysis is 100-times faster than oxidative phosphorylation [70]. A constant supply of small amounts of energy with a low oxygen consumption is advantageous for immediate energy supply [71,72]. Besides that, the function of mitochondria is considerably limited in affected cells; therefore, anaerobic glycolysis is the major mechanism of energy production. Since cancer cells are capable of increased proliferation even under these restricted conditions, we can consider that preference for anaerobic glycolysis is beneficial for cells exposed to hypoxia [66,73]. According to the above, we can suppose that cardiomyocytes from diabetic myocardium could be used to describe metabolic processes such as those of cancer cells [63]. Diabetic myocardium is characterized by a state of pseudohypoxia, as a result of electron transport chain damage associated with a limitation of oxidative phosphorylation and impairment of HIF-1 activation [32]. Pseudohypoxia is described as impaired cellular oxygen utilization capacity due to reduced levels of NAD, which may cause NADH accumulation with NADH/NAD redox imbalances [74,75]. Therefore, anaerobic glycolysis could also be advantageous for the cells of diabetic organisms [34,76]. Despite the known side effects of PDH inhibition, such as diabetes mellitus [58,77,78,79], metabolic syndrome [80], heart failure [81], and fatty liver [82], we can assume that cells of diabetic hearts use similar adaptation mechanisms to increase their survival. A sufficient supply of glucose ensures a prompt and continuous production of energy. These facts explain the advantage of anaerobic glycolysis in a diabetic heart [52]. Another important factor is age, which leads to dysregulation of molecular pathways linked to mitochondria. Increased apoptosis, declined autophagy, increased disruption of mPTP, and worsened injury after hypoxic-ischemic insults are the results of aging. The age related decrease in NAD+ contributes to substrate starvation leading to a pseudohypoxic state [83].

## 4. Metabolic Preconditioning

Adaptation of the heart to altered metabolic conditions allows the maintenance of its function. It allows the heart to meet the requirements of the body effectively [84]. One of the most commonly used experimental models for the induction of metabolic preconditioning (MPC) is streptozotocin-induced diabetes mellitus with positive structural and metabolic changes present during its acute phase [84,85,86,87,88]. The acute stage of diabetes is characterized by the inhibition of insulin secretion and decreased signaling of insulin receptors in target cells [89]. After the seven days following streptozotocin administration, changes induced by diabetes mellitus are fully developed, without side complication characteristics for the chronic stage of the disease [90]. The acute phase of diabetes mellitus persists for the next three weeks [89].

The chronic phase of streptozotocin diabetes mellitus can be induced by a longer administration of streptozotocin, i.e., for more than 60 days. It is characterized by many complications, such as neuropathy, retinopathy, nephropathy, microangiopathy, etc. [91].

Despite the fact that diabetes mellitus causes extensive changes in the structure and function of cardiomyocytes, its impact is not necessarily entirely harmful. A short-term exposure of the heart to a high glucose concentration or diabetes mellitus has proven beneficial effects against ischemic insult [92,93]. The first evidence of a compensation effect due to diabetes was presented by a study pointing out a better recovery of contractility of a diabetic heart after an I/R injury [94].

The acute phase of MPC induced by diabetes is characterized not only by metabolic changes [84], but also by positively affecting the heart efficiency and its sensitivity to pathological stimuli, remodeling the cardiomyocyte membrane, as well as cardiac mitochondria [95,96]. Indeed, remodeling has a central role in maintaining or repairing the heart tissue [97].

## 5. Involvement of Mitochondrial Connexin 43 in Cardioprotection

It has been well established that the modulation of membrane channel protein “connexin 43” (Cx43), the most abundant connexin in the heart [98], could have various cardioprotective effects [99,100,101,102]. The association of six subunits of Cx43 results in the formation of hemichannel “connexon” [103]. After transportation in secretory vesicles to the plasma membrane, two opposing connexons from adjacent cells create the “Cx channel”. Thousands of Cx43 channels aggregate into gap junction plaques at the intercalated disks [104,105]. This direct connection between two adjacent cells provides electrical and metabolic cell-to-cell coupling [106]. Cx43 hemichannels are not only precursors for Cx43 channels, but can also exist as non-junctional hemichannels at the plasma membrane and can contribute to volume regulation, to the release of ATP and NAD+ from the cytosol, and the activation of cell survival pathways [107]. In addition to predominantly localized Cx43 at the intercalated disks, 4% Cx43 is present in mitochondria due to translocation from cardiomyocytes [108,109].

MtCx43 in the cardiomyocytes is situated in the inner mitochondrial membrane (IMM) of subsarcolemmal mitochondria (SSM) where it forms an mtCx43 hemichannel [110,111]. MtCx43 import to the IMM of SSM is mediated by the interaction between Cx43 with the heat shock protein 90 (HSP90) and translocase of the outer membrane 20 [109]. The physiological role of mtCx43 is not fully clarified, but some studies support its involvement in the regulation of K^+^ fluxes [112], mitochondrial respiration [113], oxygen consumption [111,112], mitochondrial redox state [114], and in mitochondrial Ca^2+^ homeostasis [115] (Figure 1). In this context, it is understandable that mtCx43 is attributed to cardioprotection.

The implication of mtCx43 in the cardioprotective pathway of IPC has been mostly elucidated [109,111,116]. The protein level of mtCx43 very rapidly increased in response to IPC and was maintained for at least 90 min in a pig model of IPC [108], whereas attenuation of mtCx43 was associated with lost IPC cardioprotection [108]. Evidence that mtCx43 is implicated in this mechanism was also demonstrated in an experiment by Heinzel et al. in 2015, where pharmacological preconditioning by diazoxide mediated by gating of mitochondrial ATP sensitive potassium channels (KATP) and protein kinase C activation [117] was repealed in cardiomyocytes isolated from mice with a reduced Cx43 level. Mitochondrial KATP channels have also a key role in mitochondrial physiology and potential effects on several pathological processes, thanks to their involvement in cellular energetic status by regulation of organelle volume and function [118].

Indeed, diazoxide affects the generation of ROS necessary in low amounts as trigger molecules of IPC [119]. In the experimental model of Cx43-deficient mice, nitric oxide (NO) production was significantly lower compared to wild type control mice [120]. Increased S-nitrosation of mtCx43 by IPC elevated mitochondrial permeability and subsequently ROS formation [121]. Moreover, diazoxide modulates the opening of the mPTP [122]. The relationship between mtCx43 and mPTP has been elucidated. In this study, pharmacological inhibition of mtCx43 induced opening of mPTP in SSM by increased levels of Ca^2+^ [123]. Another study with IPC abolishment in which reduction of mtCx43 was induced by geldanamycin (which prevents translocation of Cx43 to mitochondria by blocking the HSP90 dependent pathway) confirmed that only mtCx43 is implicated in this cardioprotection [124]. MtCx43 can also be implicated in cardioprotection by interaction with proteins related to mitochondrial fraction and metabolism. MtCx43 interacts with the apoptosis inducing factor (AIF) involved in oxidative phosphorylation and redox control. Interestingly, AIF deficient mice had the same pattern of changes in ROS generation and mitochondrial complex 1 activity as Cx43 deficient mice [113]. In cardiomyocytes with overexpression of Cx43, only complex I respiration was increased, while complex II remained unchanged [113]. A close relationship between mtCx43 and with anti- and pro-apoptosis markers Bcl-2 and Bax was observed. In this experiment, elevated levels of mtCx43 were accompanied by the upregulation of Bcl-2 and inhibition of Bax in the cardiac mitochondria after hypoxic postconditioning [125].

## 6. The Role of Cardiolipin in Heart Mitochondrial Signaling

Cardiolipin, as a unique phospholipid, is an important component of the IMM [126,127]. It is a relevant indicator of mitochondrial membrane fluidity damage [128]. Due to the localization of respiratory enzymes and oxidative phosphorylation in the IMM, maintaining a positive membrane fluidity remodeling is essential to ensure the bioenergetic processes of the cell [95,129,130,131] (Figure 1). The fluidity of the mitochondrial membrane is an important part of endogenous protective mechanisms, especially in pathological conditions such as diabetes mellitus [34], ischemia-reperfusion damage [132,133], and hypercholesterolemia [130]. Maintaining membrane fluidity under load conditions at the control level improves ATP transport from the mitochondrial matrix to the cytosol of cardiomyocytes [130]. The sustainability of phospholipid composition in the mitochondrial membrane results in the proper mitochondrial function and structure, phospholipid metabolism, and energy transport [134].

Changes in the composition of the cardiolipin structure, content, and acyl chain are associated with mitochondrial dysfunction in the tissues of certain pathophysiological conditions, including apoptosis [125], ischemia [135], I/R [136], in various stages of thyroid disease [137,138], diabetes mellitus [139], aging [140], and heart failure [135,141].

Cardiolipin is an oxidatively sensitive phospholipid, particularly to ROS [142], due to a high content of unsaturated fatty acids [143]. Oxidative damage of cardiolipin negatively affects the biochemical function of mitochondrial membranes [127], which is reflected in the alteration of the membrane fluidity, ion permeability, as well as the structure and function of the electron transport chain. These alterations lead to a reduced oxidative phosphorylation efficacy of mitochondria [144,145].

Cardiolipin contributes to the protein function in the IMM and maintains the integrity and flow of the electron transport chain, including anionic carriers and respiratory chain complexes [146,147]. Cardiolipin is specifically required for electron transfer in mitochondria respiratory chain complex I [136]. Respiratory complex III of the mammalian chain contains bound cardiolipin molecules that are essential for the enzyme activity [148]. ROS induced oxidative damage of cardiolipin in mitochondria may be responsible for the observed defect in the activity of complex III [149]. Similarly, complex IV contains tightly bound cardiolipin whose removal results in a change of its structure and function [150].

Today, many diseases in which mitochondrial dysfunction has been associated with cardiolipin peroxidation have been described [151]. It seems that a high concentration of Ca^2+^ has a negative impact on mitochondrial function related to the cardiolipin peroxidation. A high concentration of Ca^2+^ together with cardiolipin peroxidation participates in mPTP opening [127]. It has been suggested the cardiolipin associated with the adenine nucleotide translocator (ANT) may be the site at which Ca^2+^ binds and activates mPTP opening. Binding of Ca^2+^ to ANT surrounding cardiolipins enhances the mobility of ANT-Cys^56^, which could be a potential pathway of Ca^2+^ for induction of mPTP opening [152] (Figure 1).

The accumulation of oxidized cardiolipin in the outer mitochondrial membrane (OMM) contributes to mPTP opening, which is also accompanied by the release of cytochrome c (Cyt c) from mitochondria into the cytosol [153]. The role of cardiolipin in Cyt c releasing from mitochondria seems to be very important in the process of apoptosis [154,155].

Cardiolipin is required to maintain the proper function of ATP synthase and facilitate its rotation, which is supported by the transmembrane proton gradient [156,157]. Cardiolipin is involved in mPTP control via affecting the function of ATP synthase [158,159]. Positive mitochondrial membrane remodeling is associated with an increased membrane fluidity, as well as increased mitochondrial ATP synthase activity in streptozotocin induced pseudohypoxic acute diabetic conditions [160].

## 7. The Role of Mitochondrial Permeability Transition Pores in Signaling Processes of Cardioprotection

Substantial evidence has revealed that the mPTP are associated with the signaling pathway of cardioprotective models and seem to be an end-effector of cardioprotection [161,162]. It has been shown that the inhibition of mPTP opening not only provides a protective strategy against reperfusion injury [163], but is also a key point in cardioprotective mechanisms such as IPC or MPC [164,165,166] (Figure 1). The cardioprotective effect of RPC has been associated with the inhibition of mPTP formation [44]. Transient mPTP opening, which allows the release of Ca^2+^ from the mitochondria into the matrix, appears to be a key mechanism in MPC [167].

Under physiological conditions, the mPTP are closed or not present. Their opening is associated with postischemic reperfusion, when the perturbations in intracellular Ca^2+^ homeostasis, ROS accumulation, and a reduction of mitochondrial membrane potential (Δψ) are characteristic [168,169]. The massive opening of mPTP results in an increase in IMM permeability and the entry of metabolites into the mitochondrial matrix, which leads to mitochondrial swelling, collapse of Δψ, reduction in the efficiency of ATP production by uncoupling the electron transport system from oxidative phosphorylation [170,171,172], and cell death [10,168]. mPTP remain closed due to low intracellular pH (˂7.0) during ischemia, but they are opened during the first minutes of reperfusion associated with the normalization of pH, which causes irreversible heart damage [162,173,174]. Although mPTP are associated with mitochondrial damage and cell death, transient mPTP opening represents one of the physiological processes that is used in the mitochondria of healthy cells [170]. In the heart, transient mPTP opening during preconditioning could be a protective tool that ensures a physiological role during damage [175]. It is believed that transient mPTP opening releases Ca^2+^ from the mitochondrial matrix to maintain mitochondrial homeostasis. Transient mPTP opening is also associated with a temporary increase in ROS as signaling molecules [176].

Increased mPTP production has also been reported in the experimental model of acute diabetes mellitus [177]. The increased formation of mPTP is presented as a compensating mechanism that facilitates the transfer of ATP molecules from the mitochondria into the cytosol, where energy supply is currently needed. Residual mitochondrial ATP production due to its increased cytosolic transfer has been shown to be adequate to maintain sufficient levels of adenine nucleotides in acute diabetic myocardium [34]. The inhibition of mPTP opening may also be achieved by pharmacological drugs. The development of inhibitors, except of a prototype compound such as cyclosporine A (CsA), is limited by side effects and a low therapeutic efficacy [178]. Similarly, there is evidence of a protective mechanism of mPTP inhibition against cancer cell survival and proliferation. mPTP has become a promising strategy for improving cancer therapies [179].

In studies by Heather et al., SSM was adapted to hypoxic conditions and thus mitochondria acquired increased resistance to oxidative damage under conditions of limited oxygen supply. These hypoxia mediated changes induced functional adaptation of mitochondria to a certain dose of stress, resembling the mechanism of the preconditioning effect [180].

mPTP represent a protein complex whose molecular composition remains unexplained. The new knowledge about the structure and regulation of this mitochondrial pore comes annually. The hypothesis about the nature of mPTP suggests that their number is increasing after a conformational change in ATP synthase after binding of Ca^2+^ [181,182]. This change should lead to the opening of the hidden megachannel. It has been discussed whether the role of ATP synthase can be changed from a key energy producing enzyme to an energy dissipating channel that leads to cell death [182]. ATP synthase, together with a phosphate carrier protein (PiC) and ANT, is organized into supramolecular units called synthasomes, which increases the efficiency of ATP production [14,183]. Cyclophilin D (CypD) regulates mPTP, as well as the dynamics of the synthasome, depending on the bioenergy state of mitochondria [184]. Cardiolipin oxidation can disrupt the interactions between the components of the ATP synthasome, which can cause destabilization in this supercomplex, thereby promoting mPTP opening [185].

However, a study by Carroll et al. denied the idea of ATP synthase as the main structural component of mPTP. mPTP opening also occurred after deletion of selected ATP synthase subunits after Ca^2+^ overload. Based on these findings, the authors unlikely considered that ATP synthase and its subunits are involved in the mPTP structure [186]. Interestingly, new findings confirmed the participation of ANT in the mPTP structure. Results achieved by Karch et al. supported the idea about ANT dependent mPTP activity, which is regulated by CypD. ANT dependent mPTP is activated in response to higher mitochondrial matrix Ca^2+^ levels, which means independently of CypD [187]. Many question marks hang over the mPTP structure, again.

## 8. Unregulated Mitochondrial Permeability Transition Pore Opening

The prolonged mPTP opening results in disruption of the mitochondrial ultrastructure, halting of mitochondrial energy, and ATP synthesis, resulting in a variety of diseases, currently without successful treatment [188]. The unregulated mPTP opening and the consequent oxidative damage are considered as the major mechanisms of mitochondrial energetic dysfunction, which ultimately lead to cell death [8,189]. mPTP opening has an impact on the release of Cyt c from mitochondria, which is associated with pathophysiological situations such as I/R injury, aging, and other degenerative diseases [127]. One of the independent risk factors that may cause structural, molecular, and biochemical changes is aging. Aging increases CypD expression and its interaction with ATP synthase leading to a higher risk of mPTP opening. Thus, mPTP are important factors controlling mitochondrial function affected by aging [190]. Cardioprotective mechanisms, such as preconditioning, could be also impaired by aging and lead to defects in protective cell signaling. In a study by Griecsova et al., the efficacy of preconditioning was attenuated in mature adult rats in contrast with younger animals. Increasing age caused the decrease of heart ischemic tolerance, as well as changes in cellular expression of proteins involved in the protective signaling [191]. Age related disorders are also associated with increased ROS production and dysregulation of intracellular Ca^2+^ levels, resulting in mPTP opening [192]. mPTP are opened during reperfusion after previous ischemic injury of the heart, leading to myocardial damage [193]. Likewise, disruption of Ca^2+^ homeostasis in addition to myocardial I/R injury also occur in neurodegenerative diseases that lead to mPTP opening [194]. Chronic diseases such as diabetes or hypertension cause changes in mitochondrial bioenergetics manifested by inhibition of respiratory chain complex activity, increased proton leakage from the IMM, increased ROS production, and Ca^2+^ overload resulting in mPTP opening [8,172,195]. It has been found that the prevention of mPTP opening by mPTP inhibitors would be beneficial in a wide range of therapeutically challenging diseases. Therefore, significant effort is being made to develop mPTP specific inhibitors that would overcome the major disadvantages of CsA. Further studies are needed to progress from research to therapeutics [188,196].

## 9. Conclusions

In conclusion, here we emphasize the necessity of maintaining a proper function of cardiac mitochondria even in situations with limited oxygen supply, such as heart ischemic disease. Since it is known that mitochondria have a crucial role in the adaptation process of the heart in unfavorable conditions, the idea of a positive effect of partial oxygen deprivation was studied in new therapeutic approaches. The expression of hypoxic genes and the preference of anaerobic glycolysis associated with the regulation of mPTP are considered as the key mechanisms. The previous findings suggest that not only functional, but also structural changes in cardiac mitochondria are involved in the adaptation process. This knowledge is supported by the relation of mitochondrial membrane composition and functional properties of the heart. The composition of the cardiolipin structure, the amount of mtCx43, and the degree of mitochondrial membrane fluidity affect the formation and opening of mPTP, which is reflected in ATP synthase activity and mitochondrial survival.

## Figures and Tables

**Figure 1 cells-08-01449-f001:**
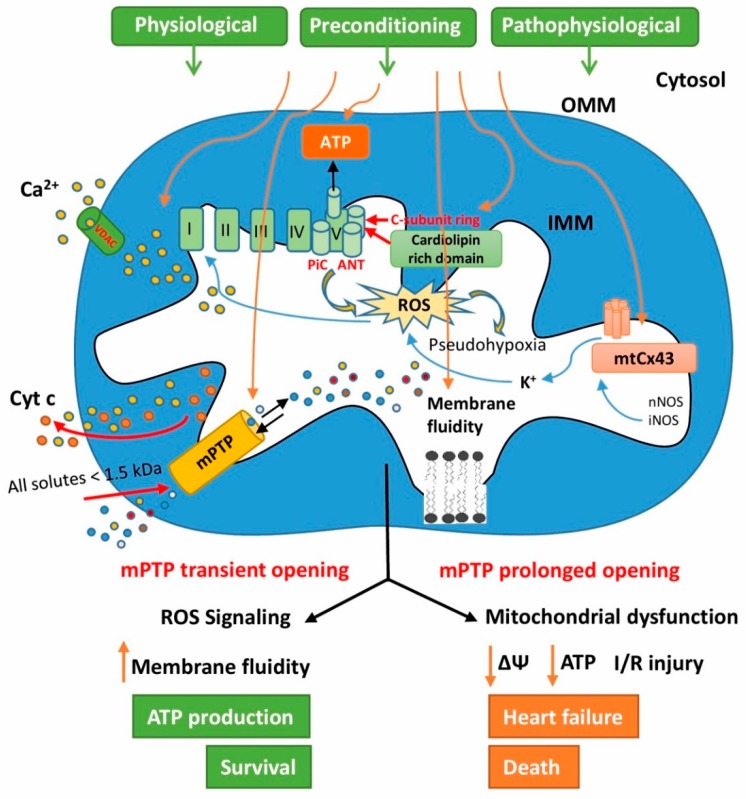
Diagram of signaling pathways affecting mitochondrial permeability transition pores (mPTP) regulation in preconditioned and pathological myocardium. For details, see the text.

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
