# Peer review of "Myocardial Adaptation in Pseudohypoxia: Signaling and Regulation of mPTP via Mitochondrial Connexin 43 and Cardiolipin"

_cells, 2019, doi:10.3390/cells8111449_

Round 1

Reviewer 1 Report

Revisions (based on comments of both reviewers) have significantly improved the manuscript. I have no further comment.

Reviewer 2 Report

The authors have satisfactorily addressed all concerns.

Reviewer 3 Report

This version has been improved after the reviewers suggestions; it is clearer and shows more recent articles that have enhanced the quality of the manuscript.

Despite the englis sintax and grammar have been signifincatly improved, some  minor grammatical mistakes may be found throughout the text, which should therefore be carefully checked again . see for exsample “ remodelation” instead “remodeling”…. 

This manuscript is a resubmission of an earlier submission. The following is a list of the peer review reports and author responses from that submission.

Round 1

Reviewer 1 Report

The purpose of this Review is to provide an overview of the myocardial adaptation under conditions of pseudohypoxia with particular attention to the signaling and regulation of mPTP via mitochondrial Cx43 and cardiololipin.
The main criticism is the lack of originality and novel perspectives necessary to promote the scientific discussion and encourage the advancement of the knowledge. All the described mechanisms and topics have been widely discussed in previous reviews and papers; in fact, the vast majority of the references date back to more than 10 years ago. Additionally, many English syntax and grammar mistakes undermine the readability of the text.

Reviewer 2 Report

In this work, Ferko and colleagues elaborate on potential signaling pathways as possible targets for cardioprotection upon partial oxygen depletion (pseudohypoxia).One such target for cardioprotection is proposed to be -as extensively addressed- the mitochondrial permeability transition pore. While some take-home messages are clear in the manuscript. This reviewer believes the manuscript is somewhat short and could even be considered still a draft. Furthermore, the authors are missing citing key studies that could potentially enhance the quality of the manuscript.

Concerns

1) Please add a list of abbreviations used throughout the manuscript.

2) I have a few change suggestions for the abstract:

"...mitochondrial permeability transition pores (mPTPs) as well as factors affecting their regulation and potentially providing..."

"The individual cardioprotective mechanisms can be interconnected to mitochondrial oxidativephosphorylation..."

"...pathways leading to mitochondrial energy maintenance during partial oxygen deprivation". 

maintenance of mitochondria during partial oxygen deprivation.

3) Line 40, change "a massive lack of oxygen" to "oxygen-limiting conditions".

4) Line 40 change "concentration of mitochondria" to "mitochondrial biogenesis".

5) Line 51, delete "energy molecules".

6) Line 57, delete "civilization".

7) Modify lines 58 to 63 as follows: 

"Partial (hypoxia) or complete(anoxia) absence of oxygen or the inability to use available oxygen due to damage of the mitochondrial respiratory chain  (pseudohypoxia) well characterizes diabetes [26] and changes several biochemical and metabolic processes [24]. Therefore, attention is required to develop new therapeutic approaches directed to mitochondria as target organelles triggering cardioprotection".

8) Line 74, change to "impulses on specific organs or tissues known as remote"

9) Please note that on line 88, an ATP/O ratio of 5.2 (31/6) for glucose oxidation versus 4.56 (105/23) for palmitate oxidation does not seem too different. In addition, as noted by the authors, heart mostly uses fatty acids mainly because of its readily availability as compared to glucose. This is especially relevant if we consider decreased blood circulation during oxygen deprivation. 

10) Please consider rewording paragraph from lines 95 to 103.

"Increasing oxidation of fatty acids in the heart reduces oxidation glucose and vice versa. The oxidation of fatty acids increases NADH and Acetyl-CoA levels, which inhibits pyruvate dehydrogenase (PDH) associated with glucose metabolism reduction [45,46]. The process of mutual regulation of glucose and fatty acid metabolism is called the Radle cycle [47]. However, the predominance of fatty acid oxidation during reperfusion versus glucose oxidation negatively affects the activity of the heart [41,43]. Consequently, manipulating heart metabolism to redirect fatty acid oxidation during reperfusion to glucose utilization may constitute a proof-of-concept on how to preserve heart function after ischemia or hypoxia [48,49].

11) On line 189, please discuss on the recently identified ATP-sensitive potassium channel in mitochondria (Nature. 2019 Aug;572(7771):609-613).

12) On line 246 please elaborate and cite on the recent article by Molkentin's group showing ANT is the mPTP.  Sci Adv. 2019 Aug 28;5(8):eaaw4597.

13) Please discard any reference on ATP synthase as being part of the mPTP and mention instead Sir John Walkers studies showing the ATP synthase story is unsustainable. Proc Natl Acad Sci U S A. 2019 Jun 25;116(26):12816-12821.

14) Most subsections are not linked and feel out of context. Please consider adding "linking" sentences at the end of each subsection so that the reader does not fell like there is a lack of continuity in the manuscript.

15) Please avoid using British or American English instinctively throughout the manuscript. Please consider professional editing services. 

Reviewer 3 Report

Ferko et al present a review of literature regarding the mPTP and cardioprotection.

The review, as it stands, doesn’t add considerably to the existing pool of mPTP reviews. Nonetheless, it presents as a cohesive and concise summary of the existing literature. Importantly, the review presents a clear hypothesis that is well supported by the sourced literature.

Major: The review extensively refers to diabetes, and this raises more important questions. How do certain pathologies impact the mPTP? Considerable literature supports the hypothesis that ischemic tolerance is reduced with certain pathologies. Is the mPTP responsible, in part, for this? Do certain pathologies create a higher opening probability of the mPTP?

Similarly, IPC and RPC is often reported to be lost with age and disease. What is the role of the mPTP in this? Is it simply a higher opening probability? Is upstream signalling impaired?

Further commentary regarding cardioprotection, age/disease and mPTP would be of great benefit to this review. It is touched on in places in throughout the review, but would

Minor:

Manuscript is generally well-written.

Page 1, Line 18: de novo what?

While it's a matter of preference, I prefer ‘cardiac mitochondria’ over ‘heart mitochondria’

Page 5, line 225 ‘especially on the ROS’. Would ‘on the ROS’ be better as ‘from ROS’ or ‘due to ROS’ or similar.

Page 6, line 239: ‘mitochondria dysfunction’ should be ‘mitochondrial dysfuction’.